# Dose-Dependent Outcome of EBV Infection of Humanized Mice Based on Green Raji Unit (GRU) Doses

**DOI:** 10.3390/v13112184

**Published:** 2021-10-29

**Authors:** Haiwen Chen, Ling Zhong, Wanlin Zhang, Shanshan Zhang, Junping Hong, Xiang Zhou, Xinyu Zhang, Qisheng Feng, Yixin Chen, Yi-Xin Zeng, Miao Xu, Claude Krummenacher, Xiao Zhang

**Affiliations:** 1State Key Laboratory of Oncology in South China, Collaborative Innovation Center for Cancer Medicine, Guangdong Key Laboratory of Nasopharyngeal Carcinoma Diagnosis and Therapy, Sun Yat-sen University Cancer Center, Guangzhou 510060, China; chenhw@sysucc.org.cn (H.C.); zhongling@sysucc.org.cn (L.Z.); zhangwl2@sysucc.org.cn (W.Z.); zhangss@sysucc.org.cn (S.Z.); zhouxiang@sysucc.org.cn (X.Z.); zhangxy@sysucc.org.cn (X.Z.); fengqsh@sysucc.org.cn (Q.F.); zengyx@sysucc.org.cn (Y.-X.Z.); xumiao@sysucc.org.cn (M.X.); 2State Key Laboratory of Molecular Vaccinology and Molecular Diagnostics, National Institute of Diagnostics and Vaccine Development in Infectious Diseases, School of Life Sciences, Xiamen University, Xiamen 361005, China; hongjunping1992@outlook.com (J.H.); yxchen2008@xmu.edu.cn (Y.C.); 3Department of Biological Sciences, Department of Molecular and Cellular Biosciences, Rowan University, Glassboro, NJ 08028, USA

**Keywords:** EBV infection, green Raji units, humanized mouse models, CD8^+^ T cells, CD19^+^ B cells

## Abstract

Humanized mouse models are used as comprehensive small-animal models of EBV infection. Previously, infectious doses of EBV used in vivo have been determined mainly on the basis of TD_50_ (50% transforming dose), which is a time-consuming process. Here, we determined infectious doses of Akata-EBV-GFP using green Raji units (GRUs), and characterized dose-dependent effects in humanized mice. We defined two outcomes in vivo, including an infection model and a lymphoma model, following inoculation with low or high doses of Akata-EBV-GFP, respectively. Inoculation with a low dose induced primary B cells to become lymphoblastoid cell lines in vitro, and caused latent infection in humanized mice. In contrast, a high dose of Akata-EBV-GFP resulted in primary B cells death in vitro, and fatal B cell lymphomas in vivo. Following infection with high doses, the frequency of CD19^+^ B cells decreased, whereas the percentage of CD8^+^ T cells increased in peripheral blood and the spleen. At such doses, a small part of activated CD8^+^ T cells was EBV-specific CD8^+^ T cells. Thus, GRUs quantitation of Akata-EBV-GFP is an effective way to quantify infectious doses to study pathologies, immune response, and to assess (in vivo) the neutralizing activity of antibodies raised by immunization against EBV.

## 1. Introduction

Epstein-Barr virus (EBV) is a causative agent of infectious mononucleosis (IM), and is associated with a range of human diseases, including malignancies (e.g., nasopharyngeal carcinoma, gastric carcinoma, Burkitt lymphoma, and Hodgkin lymphoma) and autoimmune diseases (e.g., rheumatoid arthritis and multiple sclerosis) [1,2]. EBV infection is asymptomatic in most individuals, and the virus establishes a permanent latent infection for life. In vitro, EBV has the ability to transform human primary B cells into immortalized lymphoblastoid cell lines [3]. In immunocompetent individuals, EBV-transformed B cells are readily removed by EBV-specific cytotoxic T cells because they express several highly antigenic viral proteins, such as the latent membrane protein 1 (LMP1), the EBV nuclear antigens 3 (EBNA3s), and EBNA2 [4]. However, in immunocompromised individuals, such as those undergoing organ transplant and patients with AIDS, EBV-transformed B cells can proliferate and lead to lymphoproliferative disorders (LPDs), such as post-transplant lymphoproliferative disease (PTLD) and AIDS-associated lymphomas [4].

EBV infects only humans in nature, whereas limited animal species, such as cotton-top tamarins, can be infected with EBV under experimental conditions [5]. Although cotton-top tamarins have been used to study EBV-induced lymphomagenesis in the past, this critically endangered species cannot serve as an experimental model anymore. Recently, the development of severely immunodeficient mouse strains, such as NOD/LtSz-*scid Il2rg^−/^*^−^ (NSG), NOD/Shi*-scid Il2rg^null^* (NOG), and Balb/c Rag2^−/−^IL-2rg^−/−^ (BRG), enabled the in vivo reconstitution of functional human immune system components after transplantation with human hematopoietic stem cells (HSCs) [6,7,8]. These mice are called humanized mice, and have been instrumental to reproduce key features of viral infections that target cells of the human immune system, including human immunodeficiency virus type 1 (HIV-1) and EBV [9,10].

Humanized mouse models of EBV infection have been previously reported [11,12,13]. The outcome of EBV infection in humanized mice varies with the different infectious virus doses. Low doses of the virus tend to induce latent infection, whereas high doses of the virus cause fatal lymphomas. The infectious dose of EBV has been determined mainly on the basis of TD_50_ (50% transforming dose) of EBV [11]. However, the determination of TD_50_ is a time-consuming progress which takes an average of 4–5 weeks. Also, the determination of the infectious dose of EBV using green Raji units (GRUs) of Akata-EBV-GFP has been established, whereas the infectious dose of the virus (GRUs) was not consistent in different papers [12,14,15,16]. Thus, in this study, we analyzed the pathological effects of different GRU-based doses of the Akata-EBV-GFP strain in humanized mice for the first time. In addition, we found that there are EBV-specific CD8^+^ T cells in these humanized mice model following EBV infection in a dose dependent manner. The correlation of EBV doses based on GRU quantification with pathological and immunological outcomes provides an important foundation for the study of EBV in humanized mice. The characterization of these GRU-dependent outcomes led to the establishment of two models, including an infection model and a lymphoma model. Such models, based on easy GRUs quantitation, will facilitate studies of EBV pathologies and immune responses relevant to passive immune protection of neutralizing antibodies and polyclonal sera raised by vaccination against EBV in vivo [14,17].

## 2. Materials and Methods

### 2.1. Humanized Mice

The construction of the humanized mice used in this study was based on NOD.Cg-*Prkdc^em1IDMO^Il2rg^em2IDMO^* (NOD-*Prkdc*^null^ *IL2Rγ*^null^, NPI^®^), and obtained from BEIJING IDMO Co., Ltd., (Beijing, China). Mice received an intraperitoneal injection (i.p.) of 20 mg/kg clinical grade busulfan. After 48 h post-injection, the mice received a single intravenous injection of 2 × 10^4^ human CD34^+^ stem cells, which were isolated from human umbilical cord blood with over 90% purity (Beijing Novay biotech, Beijing). Eight weeks post-cell transfer, human cell engraftment (hCD45^+^, hCD3^+^ and hCD19^+^ cells) in peripheral blood of mice was verified by flow cytometry (Appendix A).

### 2.2. Virus Production and Quantification of Infectious Viral Dose

To produce Akata-EBV-GFP [18,19], CNE2-EBV cells were treated with 12-o-tetradecanoylphorbol-13-acetate (TPA, 20 ng/mL) and sodium butyrate (2.5 mM) for 12 h to induce these cells to transition from EBV latency to productive viral replication cycle. After another 3-day culture in a fresh medium, EBV-containing supernatants were collected for infection. Infectious doses of Akata-EBV-GFP were quantified by the green Raji unit (GRU) assay [20,21]. Briefly, 1 × 10^4^ Raji cells (100 μL/well) were incubated at 37 °C in 96-well cluster plates with different stocks of the virus (100 μL/well). After 2 days of culture, the percentages of GFP-positive cells in total Raji cells were analyzed by flow cytometry (Beckman). The percentage of GFP-positive cells, determined by flow cytometry, was used to count the absolute number of GFP-positive cells in each well to determine the number of GRU per milliliter of the infectious Akata-EBV-GFP stocks used to inoculate the cells.

### 2.3. EBV Infection in Humanized Mice

Nine weeks after transplant of hCD34^+^ stem cells, mice were injected intravenously (i.v.) with 100 μL of virus stocks (1500 GRUs/mL, 41,000 GRUs/mL and 85,000 GRUs/mL) corresponding to low, medium, or high doses of Akata-EBV-GFP equivalent to 150, 4100, and 8500 GRUs, respectively. Mice were monitored daily for 6 weeks, during which, blood was collected weekly for analysis. Mice were humanely euthanized if they became clinically ill (e.g., weight loss of approximately 25% of their starting body weight). Six weeks post-infection, all living mice were euthanized, and the spleens, livers, and kidneys were collected for pathology analyses.

### 2.4. Establishment of Lymphoblastoid Cell Lines (LCLs) In Vitro

Fresh peripheral blood mononuclear cells (PBMCs) were isolated from the peripheral blood of one EBV-seronegative healthy donor, and 1.2 × 10^6^ human primary B cells were isolated from PBMCs. Among them, 1 × 10^5^ were infected with 100 μL of different virus stocks (1500 GRUs/mL, 41,000 GRUs/mL, and 85,000 GRUs/mL), or mock infected. After 2 h of incubation at 37 °C, primary B cells were seeded into 96-well round bottom plates at densities of 1 × 10^5^ cells/well in RF10 medium (RPMI-1640 with 10% fetal bovine serum (FBS), 5 mM HEPES buffer solution, 2 mM L-glutamine, 1 mM MEM sodium pyruvate, 100 mM MEM nonessential amino acids, 55 mM 2-mercaptoethanol, 100 mg/mL streptomycin, and 100 U/mL penicillin; purchased from Gibco (Thermo Fisher Scientific, Waltham, MA)) in three replicates per condition. Half of the culture medium was replaced once a week with fresh RF10 medium. Outgrowth was monitored by microscopy, and EBV-transformed lymphoblastoid B-cell lines were confirmed by in situ hybridization (ISH) with an EBV-encoded small RNA (EBER) probe (Zhongshan Jinqiao Bio. Co., Zhong shan, Guangdong) and flow cytometry.

### 2.5. Flow Cytometry

Beginning at two weeks post-infection, peripheral blood samples were collected to determine the immunophenotype of circulating lymphocytes using the following antibodies: hCD45-APC-Cy7 (HI30); mCD45-BV510 (30-F11); hCD19-APC (4G7); hCD3-FITC (SK7); hCD33-PE (P67.6); hCD8-PerCP-Cy5.5 (SK1); and hCD4-Pacific Blue (OKT4). Six weeks post-challenge, mice were euthanized, and the immunophenotype of splenocytes were determined using combinations of the following antibodies: hCD45-APC-Cy7; mCD45-BV510; hCD19-APC; hCD33-PE; hCD8-PerCP-Cy5.5; and hCD4-Pacific Blue. Detection of human B cells was performed using combinations of the following antibodies: hCD45-APC-Cy7; mCD45-BV510; CD19-APC, hCD38-BV650 (HB-7); and hCD24-PerCP-Cy5.5 (ML5). Detection of human T cells was performed using combinations of the following antibodies: hCD45-APC-Cy7; mCD45-BV510; hCD8-PerCP-Cy5.5; hCD137-APC (4B4-1); and hCD69-PE-Cy7 (FN50). Titration of all antibodies in this study were performed, which were purchased from Biolegend, and were used at a 1:100 dilution, unless otherwise noted [14,22]. For intracellular molecule staining, hCD8^+^hCD137^+^hCD69^+^ T cells were plated in 96-well round bottom plates, and stimulated with EBV-infected hCD19^+^ B cells for six hours in the presence of 2 µM monensin (BioLegend) and 5 µg/mL brefeldin A (BioLegend). T cells with no stimulation, and with phorbol myristate acetate (PMA)-ionomycin (Sigma) stimulation, were used as a negative control and a positive control, respectively. Following incubation, the cells were collected and subsequently surface-stained with hCD45-PE-Cy7 (2D1) and hCD8-PerCP-Cy5.5 (SK1), fixed and permeabilized with 1× Fixation Buffer/Permeabilization Wash Buffer (BioLegend), and then stained with IFN-γ-APC (4S. B3, BioLegend) and TNF-α-BV650 (MAb11, BioLegend) at a 1:50 dilution. Flow cytometry analyses were performed by CytoFLEX S flow cytometer (Beckman Coulter), and data were analyzed with CytExpert (Beckman Coulter) or FlowJo v10.5.3 (TreeStar), as described previously [14].

### 2.6. Histopathology, Immunohistochemistry, and In Situ Hybridization (ISH)

The spleens, livers, and kidneys of mice were fixed in 10% neutral buffered formalin, and embedded in paraffin. Consecutive sections were stained with hematoxylin-eosin (H&E), immunostained for the human B cell hCD20 marker, and hybridized in situ for expression of EBER, according to manufacturers’ instructions [23].

### 2.7. Quantification of viral DNA in Blood

DNA was extracted from the peripheral blood (50 μL) using a commercial DNA extraction kit (Omega). EBV DNA was quantified by a real-time quantitative polymerase chain reaction (PCR) (Roche Light Cycler 480) using a probe specific for the EBV BALF5 gene [24]. Synthetic DNA fragments of BALF5 (927–1129 bp) were cloned to puc19 vector. The plasmids identified by sequencing were used to generate a standard curve with known gene copy numbers ranging from 10^8^^–^10^−1^ copies/mL. The copy numbers of EBV DNA per ml were determined relatively to the standard curve. EBV gene expression was analyzed by reverse-transcription PCR (RT-PCR) as previously reported, using the specific primers listed in Appendix A [11].

### 2.8. Cell Sorting

hCD8^+^hCD137^+^hCD69^+^ T cells and hCD19^+^ B cells were sorted from the same spleens of mice inoculated with medium and high doses (GRUs) of Akata-EBV-GFP by MoFlo Astrios flow cytometer (Beckman Coulter). The purity of hCD8^+^hCD137^+^hCD69^+^ T cells and hCD19^+^ B cells were above 95%.

### 2.9. Statistical Analysis

Unless otherwise stated, one-way ANOVA was used to assess statistical significance. Statistical calculations were performed in GraphPad Prism 8. The sample numbers and replicates in each experiment are provided in the figure legends. *p* values less than 0.05 were considered to be statistically significant.

### 2.10. Ethics Statement

All experiments involving mice and rabbits were approved by the Institutional Animal Care and Use Committee at the Sun Yat-sen University Cancer Center (approval no. 202106), and the use of human cord blood CD34^+^ cells was approved by the Medical Ethic Committee at the People’s Hospital of Zhoushan Putuo District in Zhejiang Province (approval no. 2019KY015).

## 3. Results

### 3.1. Different Number of GRUs of Akata-EBV-GFP for the Formation of Lymphoblastoid Cell Lines In Vitro

We first explored the influence of virus doses on the outcome of EBV infection in human primary B cells by using different numbers of GRUs of Akata-EBV-GFP. Akata-EBV-GFP was generated in CNE2-EBV cells as described [18,25], and the virions were identified by transmission electron microscopy (Figure 1A). We determined the concentration of GFP-transducing virions as green Raji units (GRUs), since Akata-EBV-GFP encodes the green fluorescence protein (GFP) under the control of the SV40 enhancer and promoter. Raji B cells were infected with serial dilutions of virus stocks, and the percentage of GFP-positive cells was determined by flow cytometry, and used to calculate the absolute number of infected cells in each sample [20,21,26]. In this study, three different infectious titers of EBV (high (8.5 × 10^4^ GRUs/mL), medium (4.1 × 10^4^ GRUs/mL), and low (1.5 × 10^3^ GRUs/mL)), determined by the detection of GFP^+^ Raji B cells (Figure 1B), were used to infect human primary B cells, and to test for transformation in vitro. Microscopy observation showed clear signs of outgrowth of lymphoblastoid cells seven days after infection with low and medium doses of Akata-EBV-GFP (Figure 1C). In contrast, mock-infected primary B cells or cells infected with high doses (8.5 × 10^3^ GRUs) of the virus were almost all dead. Four to five weeks following infection with low and medium doses (GRUs) of Akata-EBV-GFP, cell outgrowth was universally EBV-positive, as detected by EBER in situ hybridization (Figure 1D). Flow cytometry analysis confirmed that cell outgrowth was CD19-positive B cells in both conditions (Figure 1E). Together, these results showed that infection of 1 × 10^5^ cells primary B cells by low (1.5 × 10^2^ GRUs) and medium (4.1 × 10^3^ GRUs) doses of Akata-EBV-GFP leads to growth transformation of B cells in vitro, whereas high doses (8.5 × 10^3^ GRUs/mL) of Akata-EBV-GFP has a lethal effect on primary B cells.

### 3.2. EBV Infection in Humanized Mice In Vivo

We further set out to establish the relationship between the number of GRUs of Akata-EBV-GFP and B-cell lymphoma in humanized mice. To this end, non-obese diabetic (NOD)-*Prkdc*^null^ *IL2R*γ^null^ mice were engrafted with human cord blood-derived CD34^+^ stem cells. We will further refer to these engrafted mice as humanized mice. Eight weeks post-transplant, about 14.96 ± 5.62% of peripheral blood mononuclear cells were hCD45^+^-positive cells (Appendix A), more than 90% hCD45^+^ cells were hCD19^+^ B cells, and a small part (<1%) were hCD3^+^ T cells (Appendix A). These results indicated that the generation of humanized mice was successful, and there was a high proportion of B cells, which is the main target cells for EBV infection, in the peripheral blood of humanized mice.

We next inoculated three different quantities of GRUs of Akata-EBV-GFP into humanized mice. Each test group consisted of a randomized five to six mice, while three control mice did not receive any virus. Mice were then monitored for survival, EBV DNA, as well as immunological parameters (Appendix A). Four to six weeks following challenge, all mice retained similar percentages of hCD45^+^ cells and hCD3^+^hCD4^+^ cells in peripheral blood (Figure 2A,B). However, mice inoculated with medium and high doses (GRUs) of Akata-EBV-GFP showed a significant decrease in the proportion of hCD19^+^ B cells (Figure 2C), concurrent with a marked increase in hCD3^+^hCD8^+^ T cells (Figure 2D). Compared with control mice, EBV DNA was detectable in the peripheral blood of mice which received medium and high doses (GRUs) of EBV, whereas most mice which got a low dose had undetectable levels of EBV DNA. Interestingly, the amount of DNA in the peripheral blood of mice infected with medium doses tends to increase over time as the percentages of hCD19^+^ B cells decrease (Figure 2E). All mice inoculated with low doses (GRUs) of Akata-EBV-GFP survived for the 6 weeks duration of the experiment. In contrast, mice that received high doses (GRUs) of Akata-EBV-GFP showed increased mortality after three weeks, and all died within five weeks, and 50% mice inoculated with medium doses (GRUs) survived the challenge for six weeks (Figure 2F).

To study the influence of different virus doses on tissues, we collected the spleens, livers, and kidneys of mice at the time of necropsy. Irregular and pale tumors were observed in spleens of mice that were inoculated with high and medium doses (GRUs) of the virus, and there was no visible difference in the spleens in the other two groups (Figure 3A). Splenomegaly was also significant in mice that received medium and high doses of Akata-EBV-GFP (Figure 3A,B). Immunohistochemical analysis showed that most hCD20-positive cells were EBER-positive cells in the spleens of mice that received medium and high doses (GRUs) of the virus, whereas only parts of hCD20-positive cells were EBER-positive cells in the spleens of mice that received low doses (GRUs) of the virus (Figure 3C). Marked infiltration of transformed lymphoid cells was also observed in the livers and kidneys (Figure 3C). The level of infiltration appeared to be related to the dose of virus inoculum (Figure 3D). The spleens of control mice were negative for EBER, and no transformed lymphoid cells were observed in the livers and kidneys. RT-PCR analysis of the spleens from control mice or mice inoculated with a low dose (GRUs) of the virus, or tumors obtained from mice inoculated with high and medium doses (GRUs) of the virus showed expression of EBNA1, EBNA2, LMP1, LMP2A, and EBER, consistent with the latency III gene expression program (Appendix A, Figure 3E). We also identified the transcripts from lytic-cycle genes, including immediate-early gene BZLF1, early gene BMLF1, and late gene BLLF1 (Figure 3F).

We also analyzed splenic lymphocytes at the study endpoint for mice euthanized 6 weeks post EBV challenge. Compared to the control group and mice that received low doses (GRUs) of the virus, the proportions of hCD45^+^ cells were increased in mice from the groups infected with medium and high doses (GRUs) of the virus (Figure 4A), whereas all mice retained a similar percentage of hCD45^+^hCD4^+^ cells (Figure 4B) and hCD33^+^ myeloid cells (Appendix A). Mice inoculated with medium and high doses (GRUs) of the virus showed a decrease in hCD45^+^hCD19^+^ cells (Figure 4C). Concurrent with the decline of hCD45^+^hCD19^+^ cells in mice that received medium and high doses (GRUs) of the virus, there was a significant increase in the percentage of hCD45^+^hCD8^+^ cells (Figure 4D).

It has been shown that the percentage of CD24^-^CD38^high^ cells was significantly higher in high EBV patients and humanized mice inoculated with 3.3 × 10^4^ GRUs of Akata-EBV-GFP when compared with healthy controls or control group mice [14,27]. Our results also showed that the hCD24^-^hCD38^high^ population was significantly expanded in the spleens of mice inoculated with medium and high doses (GRUs) of Akata-EBV-GFP when compared with the control group and mice that received low doses (GRUs) of the virus (Figure 5A). The percentage of CD8^+^ T cells tended to increase with the dose of the virus, thus, we next evaluated the percentage of activated hCD8^+^ T cells in different groups. Interestingly, there was a significant increase in the percentage of activated hCD8^+^ T cells in in the spleens of mice infected with medium and high doses (GRUs) of the virus (Figure 5B). We further explored whether the activated hCD8^+^ T cells (hCD69^+^hCD137^+^ T cell) were specific for EBV. We first isolated the hCD8^+^hCD69^+^hCD137^+^ T cells and hCD19^+^ B cells from the same spleens of mice infected with medium and high doses (GRUs) of the virus. The hCD8^+^hCD69^+^hCD137^+^ T cells were stimulated with EBV-infected hCD19^+^ B cells, and the frequencies of IFN-γ- or TNF-α-producing CD8^+^ T cells were determined by intracellular cytokine staining (ICS). EBV-infected hCD19^+^ B cells stimulated more than 4% (background-subtracted mean value) of the hCD8^+^hCD69^+^hCD137^+^ T cells to secrete IFN-γ or TNF-α (Figure 5B,C). These results indicate that the hCD8^+^ T cells are getting activated by human B cells following EBV infection in a dose-dependent manner, and a small part of hCD8^+^hCD69^+^hCD137^+^ T cells are EBV-specific CD8^+^ T cells.

## 4. Discussion

Humanized mouse models have been used as a typical model to analyze virological parameters and immune functions in vivo upon EBV infection [9,11]. One of the limitations of these studies was that the dose of the infected virus was determined mainly on the basis of TD_50_ (50% transforming dose) of EBV, which is a time-consuming quantification. As an alternate, GRUs quantitation was established as a tool to study EBV infection in vitro and in vivo [15,20,28]. For instance, Hellebrand et al. used GRUs to quantify EBV vectors expressing cytokines in order to assess EBV-mediated gene transfer into human B cells [15]. In humanized mice, GRU-quantified EBV have been used to study PD1- blockade on the development of lymphoma, neutralization by monoclonal antibodies, and the effect of CAR-T cells against lymphoproliferation of EBV+ cells [12,14,21,29]. However, the pathological effects and direct immunological responses to GRU-quantified EBV in humanized mice remained unclear [16]. In particular, the dose-dependent outcome of infection and T-cell response elicited by Ataka-GFP-EBV required characterization. In addressing these points, we found that inoculation with high doses of the virus equivalent to 8.5 × 10^3^ GRUs caused fatal B cell lymphoma, whereas inoculation with only 1.5 × 10^2^ GRUs resulted in latent infection. A decrease in the frequency of hCD19^+^ B cells, and an increase in percentage of total hCD8^+^ T cells were observed in the peripheral blood and spleens of mice infected with medium and high doses of the virus [14,17]. Interestingly, we also found that EBV-specific CD8^+^ T cells are induced in humanized mice following Akata-GFP-EBV infection with high (GRUs) doses.

Previous data showed that mice which received high doses of the EBV (1 × 10^3^ or 1 × 10^2^ TD_50_) developed B cells lymphoma, and all died within five to ten weeks [11]. Our results also showed that humanized mice inoculated with a high dose (8.5 × 10^3^ GRUs) of Akata EBV-GFP resulted in death within four to five weeks. Medium doses (4.1 × 10^3^ GRUs) also caused 50% of mice to die. To validate GRU quantification, and compare our data to previous TD50-based infections, we correlated GRUs with TD50 doses in an infection of human cord blood CD19^+^ B cells. The titer of the Akata EBV-GFP in 50% transforming dose (TD50) and the correlation of TD50 with GRUs were determined. High doses (GRUs) of Akata-EBV-GFP correspond to 10^3.48^ TD50, whereas medium and low doses (GRUs) of Akata-EBV-GFP correspond to 10^1.48^ and 10^−0.52^ TD50, respectively. Our data are consistent with previous observations using the TD50-quantified virus, and show that rapid GRUs quantification is a valid approach to study outcomes of EBV infection in humanized mice [11,12].

Gross observation of the spleens of mice which received 8.5 × 10^3^ GRUs of the virus showed lesions consistent with B cell lymphoma. Interestingly, we found that human primary B cells inoculated with a similarly high dose of EBV (equivalent to 8.5 × 10^3^ GRUs) died, and did not generate LCLs in vitro. The difference outcome of EBV infection in vitro and in vivo may be because there would be more of the virus per cell in vitro compared to in vivo, indicating that it is more important to test the infectious dose of EBV in the humanized mice instead of in vitro.

An increase in hCD8^+^ T cells in the blood and spleens of EBV-infected mice has been previously reported [11,14]. Moreover, these cells were able to control lymphoproliferation in vivo, since depletion of CD3^+^ T cells allowed the development of lymphoma in humanized mice, and suppressed the outgrowth of the transformed lymphoblastoid cell line ex vivo [13,16]. Here, humanized mice that received medium and high (GRUs) doses of the virus induced strong hCD8^+^ T cell responses in the peripheral blood and spleens, concurrently with a decline in the percentage of hCD19^+^ cells in the peripheral blood and spleens. These results are consistent with the possibility that human B cells infected by EBV could be recognized and killed by CD8^+^ T cells in humanized mice [11,13,17]. To address this possibility, we tested whether EBV-infected B cells isolated from mice inoculated with medium and high doses (GRUs) of Akata-EBV-GFP could stimulate hCD8^+^ T cells response. Indeed, human B cells isolated from the mice stimulated hCD8^+^hCD69^+^hCD137^+^ T cells to secrete IFN-γ or TNF-α. The identification of a proportion of this T cell subset activated in an EBV-specific manner, providing functional evidence for hCD8 T cell activity in this humanized mouse model of EBV infection at high doses. However, humanized mice that received medium and high doses (GRUs) of the virus developed fatal B cells lymphoma even though there were large amounts of hCD8^+^ T cells in the peripheral blood and spleens, which indicated that an EBV-induced CD8^+^ T cell response was not sufficient to control the occurrence and development of EBV-induced lymphoma. An increased frequency of hCD24^-^hCD38^high^ plasma blast B cells in hCD45^+^hCD19^+^ B cells may explain this phenomenon, at least partially [14,27]. Another reason may be that CD8^+^ T cells in humanized mice cannot be educated effectively, due to the lack of human thymus tissue [11]. Also, these CD8^+^ T cells may have expanded from a small fraction of T cells that were contained within the preparation of CD34^+^ cells, although, this possibility is very unlikely because of the lack of cytokines (e.g., human-derived interleukin-2 and -7) and costimulatory molecules necessary to maintain the survival and expansion of human mature T cells in mice. Currently, the underlying mechanisms of the development of EBV-specific CD8^+^ T in humanized mice infected with the EBV virus remain unclear. Interestingly, lymphoma arising after the depletion of CD3^+^ T cells contain an increased proportion of LMP1 and LMP2A-positive cells compared to lymphomas arising in the presence of T cells [16]. It remains to be seen if the active hCD8^+^hCD69^+^hCD137^+^ T cells identified in this study participate in this selection.

EBV infection in low doses (GRUs) of the virus resulted in a transient increase in EBV DNA load in the peripheral blood of a mouse, and EBER-positive B cells were observed in the spleens of mice, accompanied with infiltration of transformed lymphoid cells in the livers and kidneys. RT-PCR analysis of the spleens from mice inoculated with a low dose (GRUs) of the virus showed the gene expression of EBNA1, EBNA2, LMP1, LMP2A, and EBER1. We also detected the gene expression of BZLF1, BMRF1, and BLLF1, which may indicate B cells can induce lytic death. However, all mice inoculated with low doses of the virus retained similar percentages of hCD45^+^, hCD45^+^hCD19^+^, hCD45^+^hCD4^+^, and hCD45^+^hCD8^+^ cells in the peripheral blood and spleens compared to the control humanized mice. This type of latent EBV infection in humanized mouse models has been regarded as a model of human EBV latency [11].

In summary, we established two humanized mouse models, including an infection model and lymphoma model, by inoculation with high or low doses (GRUs) of Akata-EBV-GFP, respectively, which suggest that GRUs quantitation of Akata-EBV-GFP is a useful and practical alternative method to quantify the infectious doses of the virus for the in vivo studies of neutralizing antibodies of EBV in humanized mouse models.

## Figures and Tables

**Figure 1 viruses-13-02184-f001:**
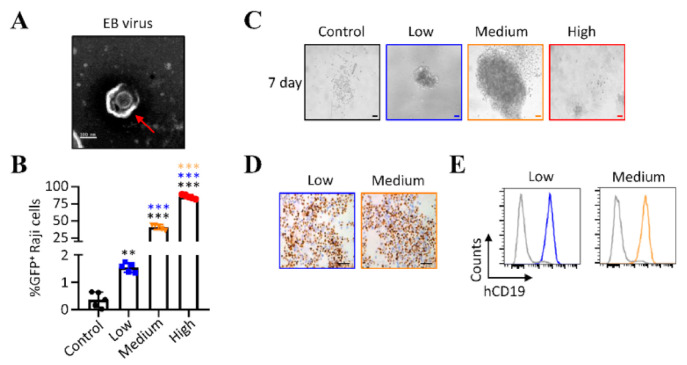
Quantitation of virus titers and infection of primary B cells. (**A**) Transmission electron microscopy (TEM) image of suspensions of EBV (magnification: 25,000 ×). Scale bar = 100 nm. (**B**) The different infectious doses of EBV were quantitated by GFP expression of Raji cells infected with defined volume of the virus. High, medium, and low represent 8.5 × 10^3^, 4 × 10^3^, and 1.5 × 10^2^ GRUs of Akata-EBV-GFP, respectively. Mean ± SD, *n* = 5, ** *p* < 0.01, *** *p* < 0.001. (**C**) Seven days after the primary B cells infected with different infectious doses of the virus (low, medium, and high), and the outgrowth detected by microscopy. Scale bar = 50 μm. (**D**) Four to five weeks after infection, the presence of EBV in the outgrowth was determined by EBV-encoded small RNA (EBER) in situ hybridization. Scale bar = 50 μm. (**E**) Four to five weeks after infection, flow cytometry analysis of this outgrowth showed CD19-positive B cells.

**Figure 2 viruses-13-02184-f002:**
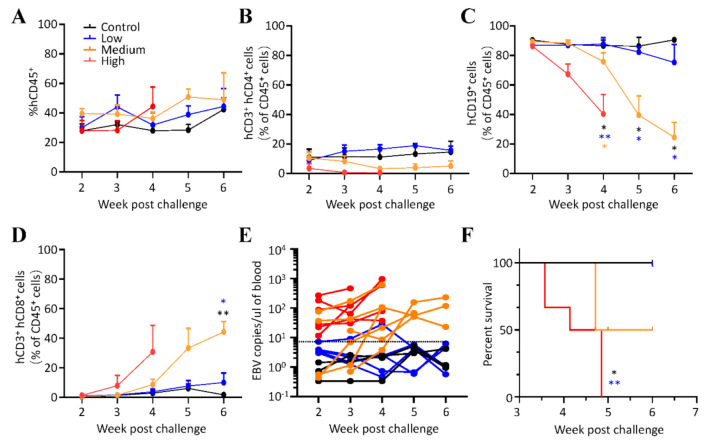
EBV infection in humanized mice. (**A**–**D**) The frequency of (**A**) hCD45^+^, (**B**) hCD3^+^hCD4^+^, (**C**) hCD19^+^, and (**D**) hCD3^+^hCD8^+^ cells in peripheral blood at the indicated time points post-challenge. hCD3^+^hCD4^+^, hCD19^+^, and hCD3^+^hCD8^+^ cells were pre-gated on the mCD45^-^hCD45^+^ human cell population. Data points represent the mean ± SEM of uninfected control mice (*n* = 3), low (*n* = 5), medium (*n* = 6), high (*n* = 6) doses (GRUs) of Akata-EBV-GFP infected mice. * *p* < 0.05, ** *p* < 0.01 (**E**) Viral DNA was quantified in the peripheral blood of uninfected control mice (*n* = 3) and mice that received low (*n* = 5), medium (*n* = 6), and high (*n* = 6) doses (GRUs) of Akata-EBV-GFP. Data are shown as the mean of three biological replicates for each mouse. Each dot represents an individual mouse, and the dotted line indicates the limit of detection. (**F**) Survival of mice was monitored weekly after infection with different infectious doses of the virus (control (*n* = 3), low (*n* = 5), medium (*n* = 6), and high (*n* = 6) doses (GRUs) of Akata-EBV).

**Figure 3 viruses-13-02184-f003:**
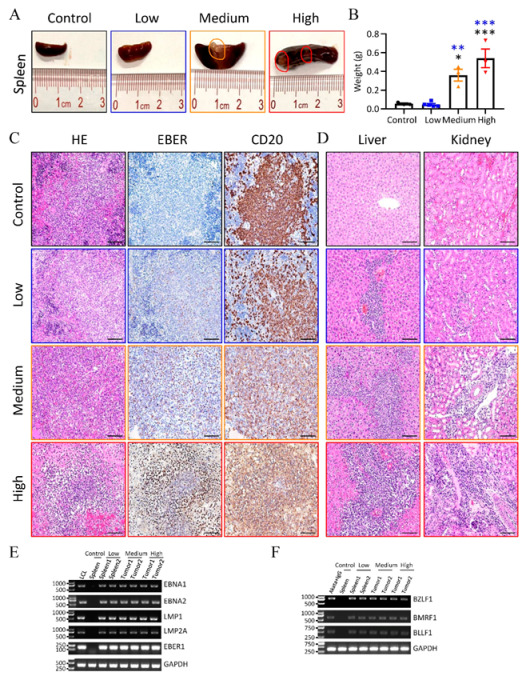
Pathology analyses of EBV-infected humanized mice. (**A**) Representative of macroscopic observation of the spleens from control mice and mice infected with increasing doses of Akata-EBV-GFP. Three mice inoculated with high doses (GRUs) of Akata-EBV-GFP became clinically ill within five weeks, and euthanasia was performed to collect the spleens, livers, and kidneys of mice. Circles indicate site of lesion. (**B**) The mean weight of spleens from (**A**). Data points represent mean ± SEM of uninfected control mice (*n* = 3), low (*n* = 5), medium (*n* = 3), high (*n* = 3) doses (GRUs) of Akata-EBV-GFP infected mice. * *p* < 0.05, ** *p* < 0.01, *** *p* < 0.001. (**C**) Splenic sections stained with hematoxylin and eosin (left), hybridized in situ for expression of EBV EBER mRNA (center), and immunostained for the human B lymphocyte marker CD20 (right). Scale bar =50 μm. (**D**) Liver and kidney sections were stained with hematoxylin and eosin (H&E). Scale bar = 50 μm. (**E**,**F**) Reverse-transcription PCR detection of latent (**E**) and lytic (**F**) EBV gene expression in the spleens or tumors from control or EBV-infected humanized mice. Spleens from two different mice inoculated with a low dose (GRUs) of the virus and tumors from two different mice infected with medium or high doses (GRUs) of the virus were examined for expression of EBNA1, EBNA2, LMP1, LMP2A, EBER1, BZLF1, BMRF1, and BLLF1. RNA isolated from the spleens of control mice (**E**,**F**) used as negative controls, and a lymphoblastoid cell line (LCL) (**E**) and anti-IgG-treated Akata-EBV cells (**F**) were used as positive controls.

**Figure 4 viruses-13-02184-f004:**
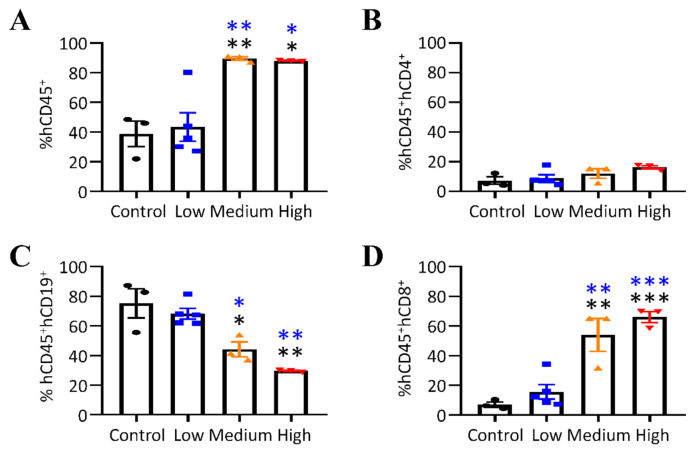
Splenic lymphocytes were analyzed in EBV-infected humanized mice. (**A**–**D**) The frequency of (**A**) hCD45^+^, (**B**) hCD45^+^hCD4^+^, (**C**) hCD45^+^hCD19^+^, and (**D**) hCD45^+^hCD8^+^ cells in spleens at the study endpoint. Data points represent mean ± SEM of uninfected control mice (*n* = 3), low (*n* = 5), medium (*n* = 3), high (*n* = 3) doses (GRUs) of Akata-EBV-GFP infected mice, * *p* < 0.05, ** *p* < 0.01, *** *p* < 0.001.

**Figure 5 viruses-13-02184-f005:**
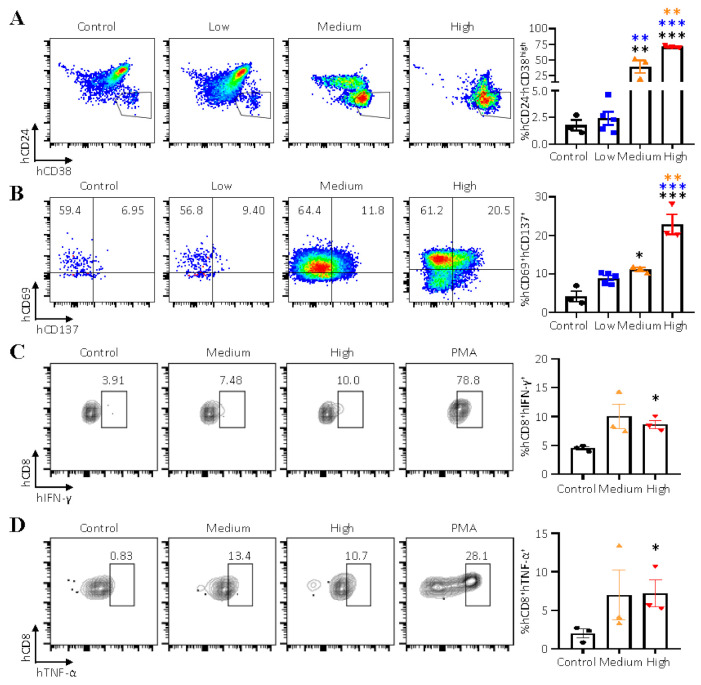
(**A**) (**left**) Representative flow plot showing hCD24^-^hCD38^high^ B cell subsets in splenocytes based upon staining with hCD38 and hCD24 (pre-gated on the mCD45^-^hCD45^+^hCD19^+^ human cell population). (**right**) The frequencies of hCD24^-^hCD38^high^ B cell subsets. (**B**) (**left**) Representative flow plot showing hCD69^+^hCD137^+^ T cell subsets in splenocytes based upon staining with hCD69 and hCD137 (pre-gated on the mCD45^-^hCD45^+^hCD8^+^ human cell population). (**right**) The frequencies of hCD69^+^hCD137^+^ T cell subsets. (**C**) (**left**) Representative flow cytometric analysis of the proportion of hIFN-γ^+^ EBV-specific hCD8^+^ T cells after coculture with EBV-infected B cells. (**right**) The frequencies of hIFN-γ^+^hCD8^+^ T cells. (**D**) (**left**) Representative flow cytometric analysis of the proportion of hTNF-α^+^ EBV-specific hCD8^+^ T cells after coculture with EBV-infected B cells. (**right**) The frequencies of hTNF-α^+^ hCD8^+^ T cell subsets. Data points represent mean ± SEM of uninfected control mice (*n* = 3), low (*n* = 5), medium (*n* = 3), high (*n* = 3) doses (GRUs) of Akata-EBV-GFP infected mice, * *p* < 0.05, ** *p* < 0.01, *** *p* < 0.001.

## Data Availability

The data presented in this study are available on request from the corresponding author.

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
