# Peer review of "Dose-Dependent Outcome of EBV Infection of Humanized Mice Based on Green Raji Unit (GRU) Doses"

_viruses, 2021, doi:10.3390/v13112184_

Round 1

Reviewer 1 Report

please find my comments in the documents attached.

Reviewer 2 Report

In this study, the authors have determined infectious doses of Akata-EBV-GFP using green Raji units (GRUs). They then infected the humanized mice with high or low doses GRUs of Akata-EBV- GFP. A high dose of EBV caused B cell lymphoproliferative disease (LPD) while a low dose of EBV resulted in a persistent latent infection. The activation of CD8+ T cells was also shown after interacting with EBV-infected B cells.

However, as mentioned by the authors, the same outcomes of infection of high and low dose of EBV to humanized mice have been reported in Yajima M et al. (J Infect Dis, 2008).

The authors emphasized in the abstract “Previously, infectious doses of EBV used in vivo have been determined mainly on the basis of TD50 (50% transforming dose), which is a time-consuming process and requires large amounts of human primary B cells.” However, if the virus itself is not a recombinant one with GFP inserted into the genome, it is not possible to use the GRU for quantifying the virus dose.

Moreover, for virus titration using TD50, it only requires plating of B cells at the density of 2 × 105 cells per well in 6-well plates and then inoculation with serial 10-fold dilutions of virus preparation. Whether this is regarded as a large amount of B cells is quite subjective.

In recent papers, GRU has been used to quantify the EBV dose before infecting humanized mice:

  1. Valery Volk et al. PD-1 Blockade Aggravates Epstein-Barr Virus+ Post- 451 Transplant Lymphoproliferative Disorder in Humanized Mice Resulting in Central Nervous System Involvement and CD4+ T 452 Cell Dysregulations. Front. Oncol., 2021.
  2. Constanze Slabik et al. CAR-T Cells Targeting Epstein-Barr Virus gp350 Validated in a Humanized Mouse Model of EBV Infection and Lymphoproliferative Disease. Mol Ther Oncolytics. 2020.

For all the above reasons, the authors should have a better justification for publishing this work. At least, they should also determine the TD50 dosing curve and find out a convertible scale to GRU.

Round 2

Reviewer 2 Report

The authors have addressed my review comments.

This manuscript is a resubmission of an earlier submission. The following is a list of the peer review reports and author responses from that submission.

Round 1

Reviewer 1 Report

In this manuscript, Chen et al. have investigated how viral dose impacts outcomes of infection by the Akata strain of EBV in a humanized mouse model.  The authors first demonstrate that administering low or medium doses of virus leads to transformation of human B cells in vitro, whereas a high dose leads to cell death.  The authors then go on to show viral dose-dependent effects on viral copy number, survival time, and the frequency of CD19+ and CD3+ cells within the human immune cell compartment in vivo.  Increasing viral dose also appeared to be associated with an increased frequency of T cells expressing CD69 and 4-1BB, suggesting an activated status.  Perhaps most interestingly, the authors provide data suggesting that viral dose is positively correlated with the appearance of CD19+ cells lacking CD24 and expressing high levels of CD38, suggesting a plasma blast or plasma cell status.  While the results presented here do suggest the potential to gain new insights from assessing how viral dose affects outcomes in vivo, in their current form the studies presented here are too limited to allow meaningful conclusions.

Major Concerns:

  1. The origin of the T cells present in this humanized mouse model is unclear. Are these T cells that developed from transplanted CD34+ HSPCs? In this case, the T cells would have been selected in the murine thymus and it is not clear that they would be able to respond efficiently to EBV peptides presented by human HLA molecules.  Alternatively, they may have expanded from a small fraction of T cells that contained within the preparation of CD34+ cells, since the methods state these were isolated from umbilical cord blood with "over 90% purity".  In this case, the T cells are expected to be able to recognize antigens presented by HLA molecules on autologous human B cells, but depending on the size of the starting population the TCR repertoire may be highly restricted so it is not clear that a sufficient starting diversity of TCR clonotypes is present to enable efficient recognition of viral peptides. 

  1. Plots in Figures 2 and 4 only show T cell/B cell marker frequencies within the human compartment. The significance of this is difficult to interpret, since the frequency of T cells and B cells within the human compartment will tend to be reciprocal in this model. It would be much more meaningful to determine whether there has been numerical expansion of each of the populations of interest.

  1. Figure 2C shows a viral-dose associated reduction in the frequency of circulating CD19+ cells; Figure 3C shows viral-dose dependent levels of EBER staining that co-localize with CD20+ cells in spleen; Figure 5A shows apparent transition of CD19+ cells in spleen to a plasma blast or plasma cell phenotype (presumably CD20-negative). While these results are potentially interesting, they raise a number of questions. Is the apparent reduction in frequency of circulating CD19+ cells (Fig 2C) due to virally induced lytic death of B cells, or to viral antigen-dependent killing by T cells, or to transition to a plasma cell phenotype (which can be CD19-negative)?  It is difficult to reconcile these results due to the different detection methodologies in each case, and difficult to interpret what may be happening in regards to changes to B cell differentiation state based on the small number of markers used for analysis.

  1. It is not clear whether the activated T cells that appear at medium or high viral doses in Fig 5B are specific for EBV, or whether this cell surface phenotype appears as a result of some other process in these mice (e.g. inflammation or destruction of murine tissues associated with their imminent death at these viral doses).

  1. No analysis was performed to assess viral gene expression patterns, and so the analysis has not addressed the critical question of whether viral dose affects EBV latency state in this model.

Minor Concerns:

  • The source of the B cells used for the in vitro experiment shown in Fig. 1 is not clear. If the source was adult PBMCs (and not cord blood B cells), was the sample confirmed to be negative for EBV?
  • Error bars appear to be missing from the plots in in Fig. 2A-E, and it is not clear how many mice were used for the survival study in Fig. 2F
  • Figure 5 needs labels on the flow plots to make it clear which viral dose is represented in each plot.  It is also not clear why there are 4 plots shown, but apparently only 3 viral doses used.
  • The methods state that viral infection was performed by i.v. injection of 100 milliliters of viral stock solution. Perhaps the units here should be microliters?

Reviewer 2 Report

Please find my comments attached in the PDF file.

Haiwen Chen and colleagues submitted an article with the title “Dose-dependent outcome of EBV infection of humanized mice based on Green Raji Unit (GRU) doses“ for the special Issue “Interplay between Viruses and Host Adaptive Immunity” to the journal Viruses. They investigated the effect of Akata-EBV-GFP using green Raji 23 units (GRUs) and characterized the dose-dependent effect in CD34 reconstituted humanized mice. High dose of infection (8.5 × 103 GRUs/ml )resulted in an fatal B cell lymphoma development detected in the spleen and a depletion of B cells and an increase of CD8+ T cells in the blood and spleen of the animals. Low dose resulted in a latent infection without changing the immune cell profile.

However, EBV infection in humanized mice has been published multiple times (96 publications found in pubmed e.g., Yajima M et al. , T cell-mediated control of Epstein-Barr virus infection in humanized mice. J Infect Dis. 2009 Nov 15;200(10):1611-5). Also publications showing similar results as described here (B cell depletion and increased CD8 T cell population) have been published e.g., by Yajima and colleagues before (Yajima M et al. A new humanized mouse model of Epstein-Barr virus infection that reproduces persistent infection, lymphoproliferative disorder, and cell-mediated and humoral immune responses. J Infect Dis. 2008 Sep 1;198(5):673-82).

In addition, Akata-EBV-GFP have been described before (Hellebrand E, Mautner J, Reisbach G, Nimmerjahn F, Hallek M, Mocikat R, Hammerschmidt W. Epstein-Barr virus vector-mediated gene transfer into human B cells: potential for antitumor vaccination. Gene Ther. 2006 Jan;13(2):150-62). Moreover, GFP-EBV has also been used in humanized mice (Ma SD, Yu X, Mertz JE, Gumperz JE, Reinheim E, Zhou Y, Tang W, Burlingham WJ, Gulley ML, Kenney SC. An Epstein-Barr Virus (EBV) mutant with enhanced BZLF1 expression causes lymphomas with abortive lytic EBV infection in a humanized mouse model. J Virol. 2012 Aug;86(15):7976-87. doi: 10.1128/JVI.00770-12). None of these articles are cited and discussed in this article.

Therefore, the overall novelty is missing in this publication and there is no scientific question in the context of EBV infection or treatment. The outlook to use this model for studies of vaccine (line 326) is difficult because these humanized mouse model (CD34 in NSG or NOG) show an overall low human myeloid cell development and therefore the interaction with myeloid cells and T cells / B cells is limited (in addition to a deficiency in Ig class switch).  

Minor commends / suggestions:

- Are these CD8+ T cells able to control infected B cells ex vivo?

- Figure 2: CD3+CD4+ of CD45? Should be mentioned on the y-axes

- Figure legends in general: SEM (standard error of the mean) instead of SME (?)

- Figure 2 A, B, C, D should contain SEM or SD at each time point to show the range in between the groups

- usually the percentage of human T cells increases over time in this kind of humanized mice (starting with low percentage and finally reach equivalent levels as B cells) but this seems not to occur here

- Figure 4: The percentage of CD19 and CD8+ changed in medium and high dose infected animals but even more the overall number of cells will extremely differ (because spleens were enlarged). If possible, please include also the total number of cells, which will be even more impressive and significant!

- Figure 5: please include the information “control”, “low”, “medium” and “high” on top of the different dot plots

Material and method:

- Please include the information about the flow cytometer and the software, which was used

- A general 1:100 dilution without titration is unusual and is also not in accordance with the manufacturer’s recommendation (5 µl/100 µl staining buffer = 1:20).

- Please also include the clone of each antibody

- Ehical approval for using human samples (Cord Blood CD34) should be included